# Severe Late-Onset Grade III-IV Adverse Events under Immunotherapy: A Retrospective Study of 79 Cases

**DOI:** 10.3390/cancers13194928

**Published:** 2021-09-30

**Authors:** Jean-Matthieu L’Orphelin, Emilie Varey, Amir Khammari, Brigitte Dreno, Anne Dompmartin

**Affiliations:** 1Department of Dermatology, Caen-Normandie University Hospital, 14003 Caen, France; anne.dompmartin@gmail.com; 2Department of Dermatology, CHU Nantes, CIC 1413, CRCINA, Nantes University, 44200 Nantes, France; e.varey@chu-caen.fr (E.V.); a.khammari@chu-caen.fr (A.K.); brigitte.dreno@atlanmed.fr (B.D.)

**Keywords:** melanoma, adverse events, immunotherapy

## Abstract

**Simple Summary:**

PD-1 inhibitors (nivolumab, pembrolizumab) and anti-CTLA-4 (CD152) (ipilimumab) are widely used in metastatic melanoma. Immunotherapy leads to prolonged lymphocyte effects, which explains the cytotoxicity underlying immune-reaction-based adverse events. Most adverse events (irAEs) occur in the first cycle of treatment at a median of 40 days, but some occur later, over 2 years after immunotherapy is initiated. IrAEs of any grade have been observed in 68.2% of patients and 10% of patients experienced severe grade III/IV irAEs. Data on late-onset irAEs are lacking. We aim to investigate late-onset grade III-IV irAEs and patient characteristics in the context of anti-PD-1 antibody treatment or combination therapy in real-life settings.

**Abstract:**

**Background:** For several decades, PD-1 has been a target in malignant melanoma (MM). PD-1 inhibitors (nivolumab, pembrolizumab) and anti-CTLA-4 (CD152) (ipilimumab) have revolutionized cancer therapy. PD-1 and CTLA-4 inhibition leads to prolonged lymphocyte effects, which explains the cytotoxicity underlying immune-reaction-based adverse events (irAEs). Most irAEs occur in the first cycle of treatment at a median of 40 days. IrAEs of any grade have been observed in 68.2% of patients, with 10% of patients experiencing severe grade III/IV irAEs. Data on late-onset irAEs are lacking. **Methods:** Data on patients with advanced melanoma (*N* = 1862) from March 2016 to March 2021 were obtained from the RicMel database, a French national multicentric biobank dedicated to the follow-up of MM patients. Patients who received anti-PD-1 therapy or a combination therapy and experienced grade III-IV irAEs were selected and analyzed at 7 months, one year and two years after treatment was initiated. **Results:** Superficial spreading melanoma (SSM) and previous oncological drug administration before immunotherapy are significant risk factors for late-onset irAEs over 2 years after beginning immunotherapy in the univariate and multivariate analysis. The other parameters—sex, mutational status, association of immunotherapy (PD-1i and CTLA-4i) and overall response—were not significantly associated with late-onset irAEs. In our real-life data study, the median onset time of grade III-IV irAES was 128 days after the initiation of immune checkpoint inhibitors (ICI) therapy. **Conclusions:** Our study, using real-life data, suggests that patients with SSM and those who have received previous oncological treatments are more likely to experience late-onset grade III-IV irAES. Further multicentric studies with wider recruitment of patients should be performed to confirm our findings, potentially leading to changes in the recommended treatment for carefully monitored at-risk patients.

## 1. Introduction

Malignant melanoma (MM) develops from pigment-containing cells known as melanocytes [1] and represents 2 to 3% of cancers [2]. The incidence rate of MM increased the most among all cancers (+371%) between 1990 and 2018, with 15,513 cases in France in 2019.

Evidence suggests that a functional immune system can act in a self-perpetuating mechanism to eliminate or durably control melanoma [3,4]. Immune checkpoints regulate the outcome of lymphocyte engagement with antigen-presenting cells and tumor cells [5], downmodulating the intensity of adaptive immune responses, which provides a spectrum of potential new targets for cancer immunotherapy.

Programmed cell death 1 (PD-1) is a key inhibitory receptor expressed by activated T and B cells. PD-1 inhibits the proliferation of immune cells that downmodulate effector functions [4,6] and promote self-tolerance. [1]. PD-L1 (B7-H1) is expressed in nonlymphoid tissues [7], whereas its expression is upregulated in MM [8]. Engagement of PD-1 by PD-L1 leads to the inhibition of T-cell receptor–mediated lymphocyte proliferation and cytokine secretion [7], consequently increasing tolerance. The blockade of PD-1–PD-L1 interaction is an effective approach for MM treatment [9]. PD-1 inhibitors, such as nivolumab, pembrolizumab [10] and ipilimumab (anti-CTLA-4 or CD152), also block prototypical T-cell checkpoints and have revolutionized cancer therapy [11,12].

PD-1 and CTLA-4 inhibition lead to continued and prolonged lymphocyte effects, which explains the cytotoxicity underlying immune-reaction-based adverse events (irAEs). Most adverse events occur in the first cycle of treatment at a median of 40 days [13]. Gastrointestinal, skin and potential irAEs leading to hyperglycemia and thyroid, hepatic and musculoskeletal disorders have been observed following nivolumab and pembrolizumab therapy [14]. IrAEs of any grade have been observed in 68.2% of patients with nivolumab, whereas potentially severe grade III/IV irAEs have been found to be experienced by approximately 10% of patients [15,16].

PD-1i treatment is stopped if a one-year complete response is achieved or tumor resistance occurs. Despite licensing for long-term use, optimal treatment duration is unknown even if some studies propose to stop the treatment for melanoma patients receiving anti-PD-1 therapy who are progression-free (confirmed by positron emission tomography (PET), coupled or not with tomodensitometry (TDM) at 12 months and with normal LDH) to avoid life-changing and life-threatening immune-mediated toxicities [17]. Nevertheless, some patients receive treatment for several years and data on late-onset irAEs are lacking.

The aim of this study was to investigate severe late-onset irAEs, i.e., grade III-IV irAEs, and patient characteristics in the context of anti-PD-1 antibody treatment or combination therapy in real-life settings.

## 2. Materials and Methods

Data on patients with advanced melanoma (*N* = 1862) from March 2016 to March 2021 were obtained from the RicMel database, a French national multicentric database dedicated to the follow-up of MM patients. Patients who received intravenous anti-PD-1 therapy or a combination of PD-1 and anti-CTLA4 were selected, and the occurrence of irAEs, the number of perfusions and overall survival rates were analyzed.

The RicMel database (Clinical Trials n°. NCT03315468) gathers data from 49 participating centers in different French regions. It received ethics committee approval on February 9th, 2012 (no. 12.108) from the Independent Ethics Committee in Paris and received authorization from the French Data Protection Agency (CNIL, DR-2012-259, 28 May 2012).

The inclusion criteria were as follows:-Patients with unresectable stage IV MM;-Patients treated with anti-PD-1 or combination therapy-Patients who experienced grade III-IV irAEs with the administered IT.

RicMel is a declarative French database. All irAEs have been previously studied by local pharmacovigilance surveys to rule out the imputability of immunotherapy from the medical field.

After screening the RicMel database, complementary medical data were collected for the included patients at the Nantes University Hospital. We reported 79 grade III/IV irAEs due to PD-1i according to standard guidelines, such as the CTCAE (Common Terminology Criteria for Adverse Events). The grading of irAEs was determined by treating physicians (dermatologists and/or oncologists) and reviewed by a physician from the RicMel database. All baseline characteristics are summarized in Table 1.

Details of 79 grade III/IV irAEs are listed in Table 2.

Details of 15 grade III/IV irAEs occurring after at least one year of ICI initiation are listed in Table 3 with details on the management of irAEs and recovery time.

### Statistical Analysis

We performed a Kaplan–Meier analysis of severe grade III-IV irAEs (event of KM curve) to confirm that most of these events occurred during the first months of treatment. We initially analyzed all grade III-IV irAEs and subsequently focused on late-onset irAEs, i.e., irAEs arising after 6 months. Thus, we performed statistical analyses of irAEs at 7 months, 1 year and 2 years.

All significant parameters in the univariate analysis were further analyzed using a multivariate analysis to determine if the parameters were independent of each other.

## 3. Results

The median onset time of our 79 grade III-IV irAES was 128 days after the initiation of ICI therapy. Irrespectively of delay appearance, the most common irAEs we reported were hepatogastroenterologic (*n* = 37, 47%), dermatologic (*n* = 12, 15%) and pneumological (*n* = 5, 6%). Most of the events were grade III irAES (*n* = 70, 88%), with only 9 (12%) grade IV irAEs.

The KM curve analysis (Figure 1) revealed that most events (i.e., grade III-IV irAEs due to immunotherapy) occur before 7 months, with 75% of events occurring before 214 days (7 months). At 7 months, 1 year and 2 years, we observed 20, 15 and 4 events, respectively. [Figure 1 near here]

Four late-onset irAEs occurred over 2 years in our cohort of 79 cases:-One SSM wild-type patient experienced grade IV myocarditis that occurred with nivolumab 759 days after the initiation of immunotherapy;-One ALM NRAS patient experienced grade III pruritus with pembrolizumab that occurred 900 days after the initiation of immunotherapy;-One SSM NRAS patient experienced grade III optic neuritis with nivolumab that occurred 1249 days after the initiation of immunotherapy;-One SSM wild-type patient experienced grade IV hepatitis with nivolumab that occurred 1375 days after the initiation of immunotherapy.

All factors associated with late-onset events (7 months, 1 year, 2 years), as identified by the univariate analysis, are summarized in Table 4.

SSM was significantly associated with irAEs at 7 months (*OR* = 3.42; *p* = 0.023), 1 year (*OR* = 4.40; *p* = 0.016) and 2 years (*OR* = 5.33; *p* = 0.156). The results at 2 years are not significant due to poor outcomes. ALM was significantly associated with irAEs at 2 years (*OR* = 24.7; *p* = 0.037) and was not significantly associated with irAES at 1 year (*OR* = 4.50; *p* = 0.298) and 7 months (*OR* = 3.05; *p* = 0.438). These results were omitted from Table 3 due to the small sample size (two ALM melanoma patients).

There was a significant risk for late-onset irAEs if patients had received previous oncological treatments (irrespective of whether these treatments were immunotherapy) at 7 months (*OR* = 8.13; *p* = 0.008), 1 year (*OR* = 5.06; *p* = 0.043) and 2 years (*OR* = 1.89; *p* = 0.589).

The other parameters—sex, mutational status, association of immunotherapy (PD1i and CTLA4i) and overall response—were not significantly associated with late-onset irAEs. We then pooled patients in two groups, those under 75 years of age and those over 75 years of age. The respective ORs at 7 months, 1 year and 2 years were 0.52, 0.61 and 0.59, but these results were not significant. The results were similar when 70 years of age was used as the cutoff.

The results were not significant with an OR = 1.57 (0.19–10.6) and *p* = 0.390 at 7 months, an OR = 1.28 (0.41–3.96) and *p* = 0.670 at 1 year and an OR = 1.42 (0.56–4.34) and *p* = 0.733 at 2 years when comparing nivolumab administered at 3 mg/kg and at a flat dose of 480 mg.

All significant parameters in the univariate analysis were subsequently analyzed by a multivariate analysis. The results are summarized in Table 5.

SSM was confirmed to be a significant risk factor by the multivariate analysis (aOR = 4.46 *p* = 0.012 at 7 months and aOR = 5.23 *p* = 0.01 at 1 year). Previous oncological drug administration before immunotherapy remained a risk factor in the multivariate analysis. Due to a lack of data, no results are available for irAEs at the 2-year time point.

## 4. Discussion

Our study shows that SSM and previous oncological drug administration before immunotherapy are significant risk factors for late-onset irAEs, i.e., irAEs that arise over 2 years after beginning immunotherapy.

The Kaplan–Meier analysis revealed that the median onset time of grade III-IV irAES is 128 days after the initiation of ICI therapy; in contrast, 91 days is reported in the literature [18] to be the median onset time of all-grade irAEs. Adverse events are commonly observed in melanoma patients treated with immunotherapy, and almost 20% of patients experience grade III-IV treatment-related adverse events [4,6]. The most common AEs are hepatogastroenterologic complications (AST or ALT increased, diarrhea, colitis, hepatitis) or pneumonitis [15,19]. In our real-life data study, the notable grade III-IV adverse events observed were colitis (*n* = 15, 19%), hepatitis (*n* = 13, 16%), cholangitis (*n* = 7, 9%) and pneumonitis (*n* = 4, 5%), which is consistent with those observed in a meta-analysis of clinical trial data [19]. Some studies have reported that fewer adverse events are associated with nivolumab [20,21,22], but these results were not statistically significant.

Four late-onset irAEs occurred over 2 years in our cohort of 79 patients. One patient experienced pruritus and another experienced hepatitis, which are well-known and frequently described irAEs associated with PDis. Additionally, one patient experienced myocarditis and one experienced optic neuritis. An extensive physical examination revealed no etiology underlying the patient’s pruritus apart from immunotherapy. Neurologic or cardiac irAEs in patients treated with ICIs are uncommon (<1%) but usually severe, with high morbidity and mortality rates [23]. We identified a patient with grade III irAE optic neuritis that occurred 1249 days after the initiation of immunotherapy. There were no indications of PD1i-induced Vogt–Koyanagi–Harada disease (VKH), which consists of uveitis, optic neuritis and choroiditis [24]. There are a few cases reported in the literature of profound vision loss due to optic neuritis occurring 4 months after immunotherapy initiation [25]. We identified a patient with grade IV irAE myocarditis that occurred 759 days after the initiation of immunotherapy. Autoimmune myocarditis secondary to the immune infiltration of CD4+ T cells into the heart [26] is a rare but often fatal event associated with checkpoint-inhibitor immunotherapy. To the best of our knowledge, no other studies have focused on late-onset irAEs; however, some case reports exist regarding late-onset cardiological irAEs [27,28], late-onset pneumonitis [29] and gastroenterological events [30]. All late-onset irAEs occurred with PD1i and none with combination therapy. The combination of PD1i-CTAL4i is more likely to provide acute irAEs than late-onset irAES, especially because it is administrated only for the first four regimens (i.e., for 9 weeks). After that, patients are treated with nivolumab only.

Our study revealed that there is a significant risk for late-onset irAEs if patients have received previous oncological treatments (irrespective of whether these were immunotherapies), indicating that cumulative exposure to treatment agents increases the risk of irAEs. A significant dose-dependent increase in PD1i-associated adverse events has been reported [21,31]; however, no similar results are available comparing a 3 mg/kg nivolumab dose (*n* = 46) to a flat nivolumab dose (*n* = 18) OR = 1.42 (0.56–4.34); *p* = 0.733, probably due to an imbalance in patient recruitment between these two groups. Given the mechanism of action of immune-checkpoint inhibitors, which promote T-cell activation and enhance T-cell activity, specific adverse events that are mostly of immunologic origin typically arise. This explains why there are more late-onset irAEs with a cumulative dose of PD1i, whereas early irAEs are more dose-dependent, rather than related to the drug mechanisms.

In addition, our results indicate that there is no association between late-onset irAEs and an initial favorable response since the ODs are 1, 0.69 and 1.54 at 7 months, 1 year and 2 years, respectively. These results are in agreement with those previously reported [32], in which irAE occurrence, irrespective of early or late onset, was found to be associated with improved PFS outcomes and better OS rates. This observation was similar regarding radiologic immune-mediated adverse reactions (i.e., colitis, pneumonitis, myocarditis) [33].

Despite the immune senescence and immune system deregulation that affects antigen-presenting cells, T lymphocytes and effector molecules in the geriatric population [34], immunotherapy treatment seems to be effective and safe for patients over 80 years of age [35]. Some researchers have reported that older patients have a better response and fewer irAES than younger patients treated with PD-1 inhibitors [36]. Others have revealed that response rates and toxicity outcomes in patients treated with checkpoint inhibitors are not altered with increasing age or comorbidity [37]. The number of comorbidities and ECOG status, or WHO performance status, are variables not associated with severe irAEs [38]. In our study, we observed a trend indicating that age over 75 years could be a protective factor (OR 0.52, 0.61 and 0.59 at 7 months, 1 year and 2 years, respectively); however, the results were not significant.

Most melanomas result from the inability of a skin phototype to repair the intracellular damage induced by UV radiation. Sunburn episodes in early childhood appeared to be more strongly related to SSM than to other variants [39]. The results of a recent study showed that helioderma could be a predictive clinical biomarker of a favorable PD-1i response [40]. Our study confirmed that SSM melanoma is a risk factor for irAEs at 7 months (*OR* = 3.42; *p* = 0.023), 1 year (*OR* = 4.40; *p* = 0.016) and 2 years (*OR* = 5.33; *p* = 0.156). These results could be explained by the long horizontal growth pattern of SSM, enabling it to achieve closer and longer contact with the immune environment than other melanomas, allowing for more contact with cytotoxic lymphocytes. Tissue-resident memory CD8^+^ T cells (T_RM_ cells) promote a durable melanoma–immune system equilibrium that is confined to the epidermal layer of the skin [41]. This explains the higher incidence of irAES associated with SSM on the one hand; on the other hand, it leads to better PFS and OS outcomes among SSM patients.

As this was a monocentric study, our results are not powerful enough to make recommendations. However, our results confirm the necessity of performing a larger multicentric cohort study analyzing data from all participating centers from the national RicMel database.

## 5. Conclusions

Our study suggests that patients with SSM and those who have received previous oncological treatments are more likely to experience late-onset grade III-IV irAES. No other study using real-life data has been performed to investigate late-onset irAES. Further multicentric studies with a wider recruitment of patients should be performed to confirm our findings.

The long-term safety of PD-1i was determined to be acceptable, since we reported very few late-onset irAEs. However, the potential for grade III-IV irAEs justifies stopping immunotherapy after achieving a one-year complete response to avoid these adverse events.

It could be relevant to modify how clinicians monitor irAEs, especially in young patients with melanoma and a long treatment duration. Even after achieving a one-year complete response, these patients should be closely followed over the long-term to quickly identify and address potential adverse events.

Ongoing prospective clinical trials will further assess the impact of immunotherapy on all grade III-IV late-onset adverse effects and provide information on how to carefully monitor individual patient risk.

## Figures and Tables

**Figure 1 cancers-13-04928-f001:**
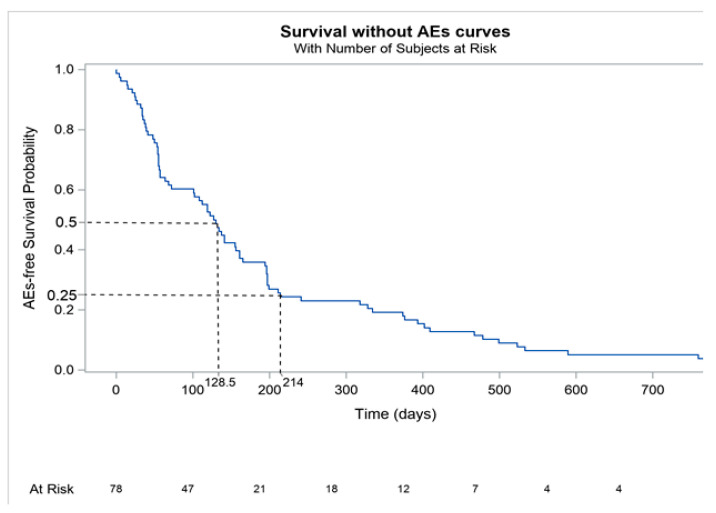
Kaplan–Meier analysis of severe grade III-IV irAEs.

**Table 1 cancers-13-04928-t001:** Patient characteristics.

Baseline Characteristics	All Cases (*n* = 79)
Age (years), mean ± std	69.5 ± 13.9
Sex, *n* (%)	
Male	41 (52)
Female	38 (48)
Type of melanoma, *n* (%) *	
NM	15 (19)
SSM	30 (38)
Ocular	4 (5)
ALM	2 (3)
LMM	5 (6)
MLM	7 (9)
Unknown primitive	7 (9)
Other	9 (11)
Mutation status, *n* (%)	
NRAS	31 (39)
BRAF	11 (14)
NRAS + CKIT	2 (3)
Wild	35 (44)
Immunotherapy, *n* (%)	
Nivo	63 (80)
Ipi + Nivo	15 (19)
Pembro	1 (1)
Line, *n* (%)	
1	30 (38)
2	24 (30)
3	16 (20)
≥4	9 (11)
Duration of actual immunotherapy (days), median (IQR)	146 (85–255)
Previously treated by PD1i, *n* (%)	29 (37)
Overall response rate (ORR), *n* (%) **	
CR Complete response	16 (25)
S Stability	12 (18)
PR Partial response	11 (17)
P Progression	26 (40)

* NM: nodular melanoma; SSM: Superficial Spreading Melanoma; Ocular: Ocular melanoma; ALM: Acro Lentiginous melanoma; LMM: Lentigo Malignant *Melanoma*; MLM: MucosalLentiginous *Melanoma*; Other: not classifiable (4) or not specified (4) or desmoplastic (1); ** Data missing for 14 patients.

**Table 2 cancers-13-04928-t002:** Characteristics and consequences of irAEs.

Characteristics	All Cases (*n* = 79)	Late-Onset AEs (≥2 Years) (*n* = 4)
Type of AEs, *n* (%)		
Cardiology	3 (4) Atrioventricular block, Myocarditis (2)	1 (25) Myocarditis
Hepatogastroenterology	37 (47) Immune colitis (15), Diarrhea (2), Cholangitis (7), Hepatitis (13)	1 (25) Hepatitis
General	5 (6) Asthenia (3), Multiple organ failure, Hypernatremia	
Hematology	2 (2) Anemia, Hyper eosinophilia	
Immunology	3 (4) Hypophysis (2), acute GvH	
Musculoskeletal system	4 (5) Myalgia, Myasthenia, Arthralgia, Elevation of CPK	
Ophthalmology	1 (1) Optic retrobulbar neuritis	1 (25) Optic retrobulbar neuritis
Dermatology	12 (15) Pruritus (3), Rash (5), Carcinoma, SJS, Bullous pemphigoid (2)	1 (25) Pruritus
Pneumology	5 (6) Immune pneumonitis (4), Pulmonary embolism	
Nephrology	3 (4) Acute renal failure, Tubulointerstitial nephritis (2)	
Neurology	4 (5) Apraxia, Demyelinating neuropathy, Neuropathy, Unknown	
Grade, *n* (%) *		
3	70 (88)	2 (50)
4	9 (12)	2 (50)
Sequelae, *n* (%) *		
Yes (except death)	12 (19)	2 (67)
Death	2 (3)	0 (0)
No	50 (78)	1 (33)

* Missing data.

**Table 3 cancers-13-04928-t003:** Focus on late-onset irAEs (>1 year, *n* = 15) of which 4 late-onset irAES >2 years.

	Type of irAES	Grade	ICI	Delay after ICI Initiation (Days)	Melanoma Characteristics (Type, Mutational Status)	Previous Systemic Treatment (Duration, Days)	Management of irAEs	Time to Recover and Sequelae
Case 1	Pruriginous rash	3	Nivo	374	MLM, BRAF	Vemurafenib-Cobimetinib (125)	Topical medication + Antihistamine + steroids	No, pruritus persistence
Case 2	Cholestasis	3	Nivo	376	SSM, wild type	IFNα (94)/Dacarbazine (350)/Ipilimumab (96)	/	Yes, grade I at 156 days
Case 3	Hepatitis	3	Nivo	393	SSM, wild type	No previous systemic treatment	Steroids per os and Cellcept (500 mgx2 per day)	No, still with steroids and hepatic disurbance when deceased
Case 4	Multiple organ failure	4	Nivo	402	SSM, NRAS	Pimasertinib (506)	Death before any treatment initiation	No recover
Case 5	Cholestasis	4	Nivo	409	SSM, wild type	Dacarbazine (192)/Nivolumab (170)	Steroids per os	Yes, at 25 days
Case 6	Immune pneumotitis	3	Nivo	467	SSM, BRAF	IFNα (534)/Dabrafenib-Tramétinib (89)/Vemurafenib-Cobimetinib (28)	/	Yes, at 44 days
Case 7	Pulmonary embolism	3	Nivo	478	NM, wild type	IFNα (117)/Ipilimumab (79)	Innohep SC 0.8 mL/day	No recover
Case 8	Apraxia	3	Nivo	499	SSM, NRAS	Pimasertinib (338)/Ipilimumab (175)	Bolus of steroids IV + Ig IV + Riuximab (4 cures)	No recover: cerebellar syndrome persistence
Case 9	Immune colitis	3	Ipi-Nivo	523	SSM, NRAS	IFNα (168)/Carboplatine-Dacarbazine (155)/Ipilimumab (156)	Steroids and remicade	Yes, at 72 days
Case 10	Hepatitis and immune colitis	3	Nivo	533	LMM, wild type	Experimental treatment (Radiotherapy + chloroquine (97))/Ipilimumab (65)	/	Yes, at 25 days
Case 11	Hepatitis	3	Nivo	589	NM, wild type	IFNα (813)/T-VEC (924)/Nivolumab (182)/Dacarbazine (88)	/	No recover
Case 12	Myocarditis	4	Nivo	759	SSM, wild type	No previous systemic treatment	Steroids IV 2 mg/kg	No, high troponine persistence
Case 13	Pruritus	3	Pembro	900	ALM, NRAS	Experimental treatment (MERCK NCT 01866319) (121)/Pembrolizumab (99)/Carboplatine-Dacarbazine (202)/Dacarabzine (338)/Carboplatine-Dacarbazine (57)	Topical medication + Antihistamine + lyrica 300 mg/days	Yes, at 42 days
Case 14	Optic retrobulbar neuritis	3	Nivo	1249	SSM, NRAS	IFNα (280)/Nivolumab (113)/Dacarbazine (109)/Nivolumab (101)/Experimental treatment (BMS 224-022 Anti-LAG3 + Nivolumab) (532))/Nivolumab (112)/Dacarbazine (140)	Bolus of steroids IV and low dose steroids basal	No recover: visual acuity reduction persistence
Case 15	Hepatitis	4	Nivo	1375	SSM, wild type	Dacarbazine (192)/Nivolumab (399)	Steroids per os	Yes, at 38 days

**Table 4 cancers-13-04928-t004:** Factors associated with late-onset events (7 months, 1 year, 2 years) by univariate analysis (*n*: number of patients with AEs at that time point; OR: odds ratio; 95% CI = l: 95% confidence interval; *p*: *p*-value for univariate logistic regression).

Characteristics	Late-Onset AEs (≥2 Years, *n* = 4)			Late-Onset AEs (≥1 Years, *n* = 15)			AEs (≥7 Months, *n* = 20)		
	OR	95% CI	*p*	OR	95% CI	*p*	OR	95% CI	*p*
Age (years)	1.03	(0.94–1.12)	0.578	0.99	(0.95–1.03)	0.545	0.97	(0.93–1.00)	0.068
Age > 75 yo	0.59	(0.06–5.98)	0.658	0.61	(0.17–2.12)	0.433	0.52	(0.17–1.63)	0.264
Male	0.29	(0.03–2.93)	0.296	0.77	(0.25–2.38)	0.653	0.90	(0.33–2.49)	0.845
*Type of melanoma*									
NM vs. others	*Not available*		0.969	0.60	(0.12–3.02)	0.539	0.69	(0.17–2.75)	0.601
SSM vs. others	5.33	(0.53–53.83)	0.156	4.40	(1.33–14.56)	0.016	3.42	(1.19–9.79)	0.023
Multiple metastatic sites	0.53	(0.05–5.33)	0.589	2.18	(0.70–6.81)	0.180	1.12	(0.40–3.17)	0.830
*Mutation Status*									
NRAS vs. others	1.42	(0.19–10.63)	0.734	0.64	(0.20–2.10)	0.464	0.91	(0.32–2.55)	0.853
Wild vs. others	1.27	(0.17–9.52)	0.815	1.57	(0.51–4.84)	0.437	0.79	(0.28–2.22)	0.655
*Immunotherapy*									
IT association	1.33	(0.13–13.74)	0.809	0.55	(0.11–2.73)	0.464	0.63	(0.16–2.47)	0.502
*Line*									
2nd, 3rd or 4th line	1.89	(0.19–19.06)	0.589	5.06	(1.05–24.26)	0.043	8.13	(1.73–38.21)	0.008
Previous IT line	1.79	(0.24–13.35)	0.576	2.34	(0.75–7.32)	0.144	2.78	(0.98–7.88)	0.054
*Overall response **									
Progressive vs. others	1.54	(0.20–11.69)	0.676	0.69	(0.21–2.32)	0.550	1.00	(0.34–2.93)	1.000

* data missing from 14 patients.

**Table 5 cancers-13-04928-t005:** Factors associated with late-onset events (7 months, 1 year, 2 years) by multivariate analyscheme 95. (*CI = l: 95% confidence interval; p: p-value for multivariate logistic regression*).

Characteristics	Late-Onset AEs (≥2 Years, *n* = 4)			Late-Onset AEs (≥1 Years, *n* = 15)			AEs (≥7 Months, *n* = 20)		
	aOR	95% CI	*p*	aOR	95% CI	*p*	aOR	95% CI	*p*
*Type of melanoma*									
**SSM vs. others**				5.23	(1.48–18.43)	0.01	4.46	(1.39–14.28)	0.012
*Line*									
**2, 3 or 4 line**				6.17	(1.21–31.54)	0.029	10.21	(2.03–51.40)	0.005

## Data Availability

The data presented in this study are available on request from the corresponding author. The data are not publicly available because they are obtained and stored from the RicMel database.

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
