# Peer review of "Severe Late-Onset Grade III-IV Adverse Events under Immunotherapy: A Retrospective Study of 79 Cases"

_cancers, 2021, doi:10.3390/cancers13194928_

Round 1

Reviewer 1 Report

1) The text should be thoroughly reviewed for style and grammatical corrections (in some cases the drug names starts with capital letter, abbreviations differs in various segments of the text, abbreviations should be properly introduced at first appearance in the text), special attention should be to the used terminology (hepatoenterogastroentrelogic)

2)...It binds with programmed cell death ligand 1 (PD-L1 [B7-H1]) and PD-L2 53 (B7-DC), expressed on antigen-presenting cells and human cancers to deliver a negative 54 signal to lymphocytes...   the binding the inhibitory ligands on tumor cells does not represent the physiological function of PD-1

3) There is no recomendations for fixed treatment duration for PD-1 inhibitors, https://www.nature.com/articles/s41416-019-0570-y, please specify

4) The methods section should be revised, the main demographic and clinical characteristics should preceed the structure of adverse events. The definition for event in KM estimates should be provided. All outcome measures should be defined in materials and methods section. What was the definition of late AEs. 

5) The results section should start with at least brief description of clinical outcomes which will provide a basis for interpretation of further results. What was the influence of patients death due to disease progression on the irAE free survival? Did median survival differed between clinical groups of patients? 

6)  P5 116-118 P7 159-161 seems like technical text, it should be removed

7) As study includes 19% of patients with nivo-ipi combination which posess higher risk of irAEs the authors should crearly describe the association between clinical groups and used treatment regimen. How the treatment structure differed between SSM and other types of melanoma in patients included in the study? 

Author Response

1) The text should be thoroughly reviewed for style and grammatical corrections (in some cases the drug names starts with capital letter, abbreviations differs in various segments of the text, abbreviations should be properly introduced at first appearance in the text), special attention should be to the used terminology (hepatoenterogastroentrelogic)

--> Integrated

2)...It binds with programmed cell death ligand 1 (PD-L1 [B7-H1]) and PD-L2 53 (B7-DC), expressed on antigen-presenting cells and human cancers to deliver a negative 54 signal to lymphocytes...   the binding the inhibitory ligands on tumor cells does not represent the physiological function of PD-1

P3L61 : Programmed cell death 1 (PD-1) is a key inhibitory receptor expressed by activated T and B cells. PD-1 inhibits the proliferation of immune cells and promotes self-tolerance . PD-L1 (B7-H1) is expressed in nonlymphoid tissues such as heart and lung whereas it is abundant in melanomas. The pro-inflammatory cytokine interferon-γ upregulates B7-H1 on the surface of tumor cell lines. Engagement of PD-1 by PD-L1 leads to the inhibition of T cell receptor–mediated lymphocyte proliferation and cytokine secretion (2) and consequently increases tolerance. The blockade of PD-1–PD-L interaction is an effective approach for melanoma treatment. 

3) There is no recomendations for fixed treatment duration for PD-1 inhibitors, https://www.nature.com/articles/s41416-019-0570-y, please specify

P3-L91 : PD-L1i treatment is stopped if a one-year complete response is achieved or tumor resistance occurs. Despite licensing for long-term use, optimal treatment duration is unknown even if some studies proposed to stop the treatment for melanoma patients receiving anti-PD-1 therapy who are progression-free  confirmed by Pet-TDM or scanner at 12 months to avoid life changing and life-threatening immune-mediated toxicities. Nevertheless, some patients receive treatment for several years and data on late-onset irAEs are lacking.

4) The methods section should be revised, the main demographic and clinical characteristics should preceed the structure of adverse events. The definition for event in KM estimates should be provided. All outcome measures should be defined in materials and methods section. What was the definition of late AEs. 

P7-L156 : We performed a Kaplan-Meier analysis of severe grade III-IV irAEs (event of KM curve) to confirm that most of these events occurred during the first months of treatment. We initially analyzed all grade III-IV irAEs and subsequently focused on late-onset irAEs, i.e., irAEs arising after 6 months.

5) The results section should start with at least brief description of clinical outcomes which will provide a basis for interpretation of further results. What was the influence of patients death due to disease progression on the irAE free survival? Did median survival differed between clinical groups of patients? 

P8-L162 : The median onset time of our 79 grade III-IV irAES is 128 days after the initiation of ICI therapy. Irrespectively of delay appearance, the most common irAEs we report are hepatogastroenterologic (n=37, 47%), dermatologic (n=12, 15%) and pneumological (n=5, 6%). Most of the events are grade III irAES (n=70, 88%) and we report only 9 (12%) grade IV irAEs.

Influence of patients death on irAES is mentioned in discussion P11-L286 : In addition, our results indicate that there is no association between late-onset irAEs and an initial favorable response since the ODs are 1, 0.69 and 1.54 at 7 months, 1 year and 2 years, respectively. These results are in agreement with those previously  reported in which irAE occurrence, irrespective of early or late onset, was found to be associated with improved PFS outcomes and better OS rates (6). This observation was similar regarding radiologic immune-mediated adverse reactions (i.e., colitis, pneumonitis, myocarditis).

6)  P5 116-118 P7 159-161 seems like technical text, it should be removed : integrated

7) As study includes 19% of patients with nivo-ipi combination which posess higher risk of irAEs the authors should crearly describe the association between clinical groups and used treatment regimen. How the treatment structure differed between SSM and other types of melanoma in patients included in the study? 

P8-L180 : Four late-onset irAEs occurred over 2 years in our cohort of 79 cases:

- 1 SSM wild patient experienced grade IV myocarditis that occurred with nivolumab 759 days after the initiation of immunotherapy;

- 1ALM NRAS patient experienced grade III pruritus with pembrolizumab that occurred 900 days after the initiation of immunotherapy;

- 1 SSM NRAS patient experienced grade III optic neuritis with nivolumab that occurred 1249 days after the initiation of immunotherapy;

- 1 SSM wild patient experienced grade IV hepatitis with nivolumab that occurred 1375 days after the initiation of immunotherapy.

P11-L271 : All late-onset irAEs occurred with PD1i and no one with combination. Combination of PD1i-CTAL4i is more likely to provide acute irAEs than late-onset irAES especially because combination is administrated only for the 4 first regimens (i.e during 9 weeks) and after that, patients are treated with nivolumab only.

 In our study, we report 22 SSM patients and 18 (81.82%) experimented toxicities with nivolumab and 4 (18.18%) with combination.  For 40 other-than-SSM melanomas, we report 30 (75%) toxicities with nivolumab and 10 (25%) with combination. 

As our recruitment is retrospective, treatment did not depend on type of melanoma.

Reviewer 2 Report

The main criticism of this study is whether the factors determining the occurrence of an increased rate of long-term side effects (SSM pathological subtype and age) have been adjusted for confounding factors such as M-category, patient's functional status or previous comorbidities.

- How did the authors demonstrate that long-term adverse effects are due to immunotherapy and not to other medications or concomitant diseases?

-Is there any relationship between late side effects and maintenance of long-term response?

-Minor comments:

Page 1, line 27: The acronym SSM appears in the abstract without having been previously mentioned. 

Page 2, line 63: "PD-1 63 and CTLA-4 inhibition lead" instead of "leads"

Page 4, lines 104-106: There are some spelling mistakes

Page 6, line 130: "SSM melanoma" should be only SSM

Author Response

The main criticism of this study is whether the factors determining the occurrence of an increased rate of long-term side effects (SSM pathological subtype and age) have been adjusted for confounding factors such as M-category, patient's functional status or previous comorbidities.

P4-L121 : Ricmel is a declarative French biobank. All irAEs have been previously studied by local pharmacovigilance survey to rule imputability of immunotherapy from medical fold.

L11-302 : …toxicity outcomes in patients treated with checkpoint inhibitors are not altered with increasing age or comorbidity (37). Number of comorbidities and ECOG status or WHO permorance status are known to be variables not associated with severe irAEs (8).

- How did the authors demonstrate that long-term adverse effects are due to immunotherapy and not to other medications or concomitant diseases?

P4-L121 : Ricmel is a declarative French biobank. All irAEs have been previously studied by local pharmacovigilance survey to rule imputability of immunotherapy from medical fold.

-Is there any relationship between late side effects and maintenance of long-term response?

Influence of patients death on irAES is mentioned in discussion P11-L286 : In addition, our results indicate that there is no association between late-onset irAEs and an initial favorable response since the ODs are 1, 0.69 and 1.54 at 7 months, 1 year and 2 years, respectively. These results are in agreement with those reported by others, in which irAE occurrence, irrespective of early or late onset, was found to be associated with improved PFS outcomes and better OS rates (6). This observation was similar regarding radiologic immune-mediated adverse reactions (i.e., colitis, pneumonitis, myocarditis) (7).

For our 4 late-onset > 2 years irAES, we are not enough powerful to conclude.

All minors comments have been integrated

Round 2

Reviewer 2 Report

Minor changes:

Page 2, line 80: Pet-TDM should be changed by positron emission tomography (PET) coupled or not with tomodensitometry (PDM)

Author Response

Thanks for review. 

Review integrated. 
